# Reanalysis of *Lactobacillus paracasei* Lbs2 Strain and Large-Scale Comparative Genomics Places Many Strains into Their Correct Taxonomic Position

**DOI:** 10.3390/microorganisms7110487

**Published:** 2019-10-25

**Authors:** Samrat Ghosh, Aditya Narayan Sarangi, Mayuri Mukherjee, Swati Bhowmick, Sucheta Tripathy

**Affiliations:** 1Structural Biology and Bioinformatics Division, CSIR-Indian Institute of Chemical Biology, 4 Raja S.C. Mullick Road, Kolkata 700032, India; samrat.ghosh2010@gmail.com (S.G.); a.n.sarangi@gmail.com (A.N.S.); mayuri1006@gmail.com (M.M.); bhowmickswati@gmail.com (S.B.); 2Academy of Scientific and Innovative Research (AcSIR), Ghaziabad 201002, India

**Keywords:** *Lactobacillus paracasei*, horizontal gene transfer, carbohydrate active enzyme, thiamin, pan/core-genome, COG, host-specific gene

## Abstract

*Lactobacillus paracasei* are diverse Gram-positive bacteria that are very closely related to *Lactobacillus casei,* belonging to the *Lactobacillus casei* group. Due to extreme genome similarities between *L. casei* and *L. paracasei*, many strains have been cross placed in the other group. We had earlier sequenced and analyzed the genome of *Lactobacillus paracasei* Lbs2, but mistakenly identified it as *L. casei*. We re-analyzed Lbs2 reads into a 2.5 MB genome that is 91.28% complete with 0.8% contamination, which is now suitably placed under *L. paracasei* based on Average Nucleotide Identity and Average Amino Acid Identity. We took 74 sequenced genomes of *L. paracasei* from GenBank with assembly sizes ranging from 2.3 to 3.3 MB and genome completeness between 88% and 100% for comparison. The pan-genome of 75 *L. paracasei* strains hold 15,945 gene families (21,5232 genes), while the core genome contained about 8.4% of the total genes (243 gene families with 18,225 genes) of pan-genome. Phylogenomic analysis based on core gene families revealed that the Lbs2 strain has a closer relationship with *L. paracasei* subsp. *tolerans* DSM20258. Finally, the in-silico analysis of the *L. paracasei* Lbs2 genome revealed an important pathway that could underpin the production of thiamin, which may contribute to the host energy metabolism.

## 1. Introduction

Among the lactic acid bacteria, *Lactobacillus* is the prominent genus, which presently embraces more than 200 species [1]. These species are commonly found in a wide variety of niches (e.g., the gastrointestinal tract, fruit, vegetables, wine, milk, and meat), including the complex microbial community. These bacteria are widely used in many biotechnological applications, e.g., as a vaccine carrier, in bioplastic production, as probiotics, or as starter cultures, indicating their high commercial value [1].

Furthermore, *L. paracasei* is remarkable in adapting to diverse habitats, especially to the gastrointestinal tract. Like other lactic acid bacteria, strains of *L. paracasei* are also extensively used as a starter culture for dairy products, in the food industry, and as probiotics [2]. The taxonomic classification of *L. paracasei* and *L. casei* has been a matter of extensive debate [2]. In the NCBI database (GenBank), several strains of *L. casei* species are misclassified, and these strains needed to be placed under the *L. paracasei* species [3,4,5]. Thus, in this study, we did extensive analysis to place the species in their respective corrected taxa.

Besides this, comparative genomic analysis among several *L. paracasei* strains reveal crucial features, such as probiotic properties and niche adaptation [2]. Notably, the function of the sortase enzyme is allied with probiotic traits, such as immune signaling and adhesion to the intestinal mucosa [6]. These enzymes are involved in the modification of cell surface proteins containing the LPXTG motif among bacteria. The number of proteins containing LPXTG motifs vary from one to more than forty per genome in many species [7]. Some reports also emphasized the role of sortase in gut bacteria where cell envelope proteins are crucial for the establishment of interaction between host and probiotic strains [8]. Interestingly, it is also noticeable that *Lactobacillus paracasei* produces bacteriocins in the gastrointestinal tract [9]. These are natural antimicrobial substances (biologically active peptides) that act against other bacteria having a specific immune mechanism [10]. In some cases, bacteriocins are viewed as an essential property for the identification of a probiotic strain [11].

The metabolism of oligosaccharides is essential for the ecological fitness of Lactobacilli strains, though there is very little knowledge of the capability of Lactobacilli to exploit oligosaccharides as a carbon source [12]. In *Lactobacillus casei,* fermentation is a crucial source of energy, which can ferment glucose, fructose, mannose, galactose, mannitol, *N*-acetylglucosamine, and tagatose [13]. It has also been reported that commensal bacteria can contribute significantly to human metabolism through vitamin B12 production. They could also synthesize vitamins B and K [14]. *Lactobacillus rhamnosus* GG, isolated from the healthy human gut is capable of incorporating vitamins like B1, B2, and B9 in the culture medium. This is the sole probiotic strain in which thiamin production has been reported [15].

Here we present a detailed analysis of genomes of the *Lactobacillus casei* group and propose corrections of 38 strains earlier identified as *L. casei* into the *L. paracasei* species after extensive genomics analysis. Our earlier assembly of the Lbs2 strain was carried out using Allpaths-LG-49856, where the GC filtration method was used to remove contaminating reads, and the genome was described as *Lactobacillus casei* Lbs2 [16]. However, analysis with CheckM revealed that the genome was only 60% complete with 1.6% contamination. We, therefore, re-assembled the genome using SPAdes assembler, followed by contamination and completeness detection using CheckM, which provided a better result (completeness: 91.28%; contamination: 0.87%) than the previous assembly. Average Nucleotide Identity (ANI) and Average Amino Acid Identity (AAI) analysis of the re-assembled genome (Lbs2) was done with 123 publicly available *L. paracasei* (*n* = 74) and *L. casei* (*n* = 49) genome assemblies, which classified our strain as a member belonging to *L. paracasei* species. Next, we performed pan-genome analysis among the *L. paracasei* strains (*n* = 75) to understand the aspects of evolution, metabolism, physiology and technological properties. Finally, an in-depth analysis revealed the technological properties of *L. paracasei* Lbs2, which may help in improving host energy metabolism.

## 2. Materials and Methods

### 2.1. Brief Description of the Key Software Used in This Study

NxTrim (Chesterford Research Park, Little Chesterford, UK): This software removes Nextera mate-pair junction adaptors from the raw reads.

SPAdes (Algorithmic Biology Laboratory, St. Petersburg, Russia): An Eulerian de Bruijn graph-based assembler, designed for the assembly of single-cell and multi-cells bacterial data sets.

CheckM (Australian Centre for Ecogenomics, Queensland, Australia): A set of tools for assessing the quality of genomes recovered from isolates, single cells, or metagenomes. It estimates genome completeness and contamination by using collocated sets of genes that are ubiquitous and single-copy within a phylogenetic lineage.

BUSCO (Swiss Institute of Bioinformatics, Geneva, Switzerland): BUSCO (Benchmarking Universal Single-Copy Orthologs) comprises of lineage-specific core Orthologs. Based on the similarity search, the quality of genome assembly and the presence of gene content is assessed.

Prokka (Monash University, Clayton, Australia): A rapid prokaryotic genome annotation pipeline.

Roary (The Wellcome Trust Sanger Institute, Hinxton, Cambridge): A high-speed standalone pan-genome pipeline, which takes annotated assemblies in GFF3 format as input (produced by Prokka) and calculates the pan-genome.

FastTree (Lawrence Berkeley National Lab, Berkeley, CA, USA): Infers approximately-maximum-likelihood phylogenetic trees from alignments of nucleotide or protein sequences.

### 2.2. Processing of Reads, Re-Assembly, and Acquisition of Publicly Available Assemblies

Illumina paired-end (length 151 bp, coverage 500×) and mate-pair (length 101bp, coverage 150×) libraries of *L. casei* Lbs2 were previously generated using the Illumina MiSeq platform [16], and were contamination-checked and re-assembled to fix the assembly issues. For this, the reads from the paired-end library (*n* = 4,784,475 × 2) were quality checked using FastQC (https://github.com/s-andrews/FastQC), which revealed all reads were of fairly good quality, without adapters and with an average length of 150 nt. Contaminant reads from the mate-pair library (*n* = 2,180,266 × 2) were purged using NxTrim [17]. Finally, cleaned paired-end along with mate-pair reads were re-assembled using SPAdes v3.11.1 genome assembler [18] with k-mers 25 to 97 with a step size of 6 and coverage cutoff as auto.

We retrieved all the genome assemblies classified as *L. paracasei* (*n* = 74) and *L. casei* (*n* = 49), (a total of 123 genomes) from the NCBI genome database (Accessed on 8 July 2018).

### 2.3. Quality Assessment of Genomes

Contigs obtained using SPAdes v3.11.1 were analyzed with CheckM v1.0.11 (Australian Centre for Ecogenomics, Queensland, Australia) [19] to evaluate the completeness and contamination using the lineage_wf workflow. Plausible contaminants were detected iteratively with reference to tetra-nucleotide distributions using the ‘outliers’ command followed by contamination detection, to get a final set of contigs with maximum completeness and minimum contamination. Completeness was also cross verified using Benchmarking Universal Single-Copy Orthologs (BUSCO) [20]. The above process of contamination and completeness detection was done for all the genomes downloaded from the NCBI database as *L. paracasei* and *L. casei* as well.

### 2.4. Average Nucleotide Identity (ANI) and Average Amino Acid Identity (AAI) Calculation

To check the genetic relatedness, we calculated ANI and AAI with all *L. paracasei* and *L. casei* genomes with the *L. paracasei* Lbs2 strain as a reference based on the “pyani” (https://github.com/widdowquinn/pyani) and “CompareM” (https://github.com/dparks1134/CompareM) package, respectively. All the results were visualized and plotted with R package “pheatmap” in R studio (https://www.rstudio.com).

### 2.5. Gene Prediction and Annotation

Coding sequences (CDSs) and all types of RNA genes of the Lbs2 strain and 74 *L. paracasei* strains were predicted and annotated with Prokka v1.13 (Monash University, Clayton, Australia) [21]. cluster of orthologous groups (COG) and pathway annotations were carried out with STRING [22] and KEGG databases [23] using BLASTP [24] keeping an e-value cut-off of <10^−5^, respectively. Furthermore, for cataloging, subsystem categories of the *L. paracasei* Lbs2 strain were annotated using the Rapid Annotation using Subsystem Technology (RAST) webserver [25].

### 2.6. Construction of Pan/Core-Genome Families and Unique Genes

GFF formatted files derived from Prokka (v1.13) were analyzed using Roary (Rapid standalone pan genomic pipeline) [26] to elucidate the Pan/core-genome and unique genes. In Roary, Markov cluster (MCL) algorithm [27] was used to cluster orthologous genes. Additionally, core-genome and singleton analysis, among *L. paracasei* Lbs2, *L. paracasei* subsp. *tolerans* DSM20258, and *L. paracasei* Lpc-37 strains were carried out with OrthoVenn (a web platform for orthologous gene clustering) [28].

### 2.7. Phylogenomic Analysis and Ka/Ks Calculation of Core Genomes

A phylogenomic tree based on core genes among the *L. paracasei* strains was built with the help of Roary and FastTree software (Lawrence Berkeley National Lab, Berkeley, CA, USA) [29]. Core gene sets were aligned using Roary based mafft [30], alignment was concatenated, and the final alignment was taken as input in the FastTree package for the construction of the maximum likelihood tree. Finally, the tree was visualized and edited using iTOL v4 [31].

The Ka/Ks value for each core gene pair was computed by the KaKs_Calculator [32] and ParaAT software (Beijing Institute of Genomics, Beijing, China) [33].

### 2.8. Comparative Genomic Analysis

Full-length genomes of *L. paracasei* Lbs2, *L. paracasei* subsp. *tolerans* DSM20258, and *L. paracasei* Lpc-37 were aligned by progressive MAUVE [34]. The circular map of *Lactobacillus paracasei* Lbs2 genome was created by the DNAPlotter software [35]. Comparison of important core-gene clusters and thiamin biosynthetic gene clusters among *L. paracasei* Lbs2, *L. paracasei* subsp. *tolerans* DSM20258, and *L. paracasei* Lpc-37 was done with the Easyfig comparison tool [36]. Stress linked genes and carbohydrate transporters were determined using the GO database (http://archive.geneontology.org). The carbohydrate activating enzyme families were detected by using the stand-alone dbCAN [37] against CAZy database [37]. In addition, sortases and proteins carrying LPXTG motifs were predicted with the stand-alone LOCP tool [38]. Furthermore, bacteriocins, CRISPRs, prophages, ISs, and GIs were identified using the BAGEL4 web-based resource [39], CRISPR finder stand-alone tool [40], PHASTER web server [41], ISsaga web platform [42,43], and IslandViewer4 [44], respectively.

### 2.9. Data Availability

Raw reads of the *Lactobacillus paracasei* Lbs2 strain are available under Bioproject PRJNA255080, and the re-assembled genome sequence of the same is present at GenBank with the accession number JPKN00000000.3.

## 3. Results and Discussion

### 3.1. Genomic Properties of the L. paracasei Strains

Pair-wise genome comparison metrics, such as Average Nucleotide Identity (ANI) and Average Amino Acid Identity (AAI) with >95% threshold as the cut-off, are frequently used as an operative method for species boundary demarcation [45,46]. In addition, it is also evidenced that the confidence of classification or probability of misclassification of a newly assembled genome depends on only ANI and AAI values from multiple genome (established species genome) comparison. In the present analysis, we calculated the ANI and AAI values for the reassembled Lbs2 genome with *L. paracasei* (*n* = 74) strains and *L. casei* (*n* = 49) strains separately. The ANI values are found to be around 98% and AAI values varied from 97% to 99% with *L. paracasei* (*n* = 74) strains (Appendix A). With *L. casei* (*n* = 49), the ANI and AAI varied from 77% to 98%, and 85% to 98%, respectively (Appendix A). Therefore, we inferred that our reassembled genome, Lbs2, should be placed under the *L. paracasei* species. Moreover, as per ANI and AAI analysis, we proposed that 38 publicly available strains of *L. casei* need to be moved out and placed under the *L. paracasei* species (Table 1).

To evaluate the degree of completeness among the *L. paracasei* genomes (*n* = 75), we used CheckM. The completeness of the reassembled *L. paracasei* Lbs2 genome is around 91.28% with 0.81% contamination compared to the earlier reported analysis with 62% completeness and 1.6% contamination (Appendix A) [16]. Among the strains studied, the genome of the *L. paracasei* Lpc-37 strain is found to be at a higher level of completeness (100%) and without any contamination (0%), as per the CheckM output (Appendix A). Because of this, the *L. paracasei* Lpc-37 strain is considered for further analysis. Genome assembly completeness based on BUSCO analysis among the *L. paracasei* strains varied from 42.7% to 99.6% (Appendix A).

The Genome sizes of *L. paracasei* strains (*n* = 75) ranged between 2.36 and 3.25 Mb, with an average value of 2.97 Mb. The GC contents varied from 46.05% to 46.97%, with a mean value of 46.30%. Additionally, N50 (base pairs) and L50 (number) values are from 2658 to 3,112,081 and 1 to 292, respectively (Appendix A). Besides these, other genomic feature details and source of the organisms are listed in Appendix A**.**

The reassembled genome of the *L. paracasei* Lbs2 strain was re-annotated with Prokka, resulting in a total of 2380 genes, including 2308 protein-coding, 20 t-RNA, 3 r-RNA, and 49 misc-RNA genes (Figure 1).

Further, annotation by the RAST server showed that a major proportion of protein coding groups are “Carbohydrates” (26.71%; *n* = 105), “DNA Metabolism” (14.50%; *n* = 57), and “Cofactors, Vitamins, Prosthetic Groups, Pigments” (12.97%; *n* = 51) (Appendix A). Notably, undigestible carbohydrates are the prime source of energy for the gut microbes. These undigested carbohydrates are originating from the plant sources that are defiant to enzymatic degradation and are not absorbed in the upper part of the intestinal tract. Such dietary compounds reach the large intestine, where they are get hydrolyzed by a limited range of organisms [47]. This could be one of the reasons for the presence of a higher percentage of carbohydrate metabolism-related genes in gut isolate *L. paracasei* Lbs2. Besides this, many reports suggested that carbohydrate metabolism in Lactobacilli is also crucial for niche adaptation or survival [48]. The distribution of functional COG categories across the *L. paracasei* strains (*n* = 75) are illustrated in a heatmap (Figure 2), and the pathway annotation of genes specific to *L. paracasei* strains are presented in Appendix A. Importantly, the presence of the higher number of ABC transporter gene family members among the *L. paracasei* strains (*n* = 75) may be responsible for the regulation of gene expression and interaction with the environment (Appendix A).

### 3.2. Pan/Core-Genome and Unique Genes Analysis

The pan/core-genome of *Lactobacillus paracasei* strains (*n* = 75) was calculated by Roary software. The pan-genome of 75 *Lactobacillus paracasei* strains hold 15,945 gene families (215,232 genes), while the core genome contained 243 gene families (75 × 243 = 18,225 genes) (Appendix A and Figure 3A). Roary pipeline uses an in-built CD-HIT clustering algorithm, which clustered the protein sequences with a sequence identity of 100% and a matching length of 100%. If one sequence is identical along its entire length to other orthologous counterparts in other species/strains, then it is said to be a core gene. The core genes include mostly ribosomal proteins and several housekeeping genes. In-addition, the COG functional annotation of these core genes showed that the majority of them belong to “translation, ribosomal structure and biogenesis (J)” and “transcription (K)” (Appendix A and Figure 3B).

Moreover, Ka/Ks ratios of all the core genes are found to be less than 1 (Figure 4 and Appendix A), suggesting strong purifying selective pressure (negative selection), which may reduce the genomic decay process.

Depending on the niches, we further analyzed the unique genes across the 75 strains of *L. parcasei*. The results showed that highest number (*n* = 251) of unique genes are associated with the Lpp126 strain while lowest number (*n* = 4) is found in the NRIC1917 strain. Furthermore, the *L. paracasei* Lpc-37 strain contained 13 unique genes. The reassembled genome, i.e., *L. paracasei* Lbs2, carries 54 genes. Interestingly, genes belonging to thiamin biosynthetic pathways (*thiE_2, thiM_2*) and the internalin gene family (*inlJ_1*) are part of the unique genes (*n* = 54) present in the Lbs2 strain (Appendix A).

### 3.3. Whole Genome Phylogenetic Analysis Reveals Closeness between *L. paracasei* Lbs2 and L. paracasei subsp. *Tolerans* DSM20258 Strain

Considering the conserved nature of the 16S rRNA sequence, it is broadly used over other functional genes for the identification of the *Lactobacillus* species [49]. The conventional mode of phylogenic tree construction from the 16S rRNA gene is usually unstable and needs the inclusion of functional genes to enhance resolution at the strain level [50]. Thus, we have built a highly robust maximum-likelihood phylogenetic tree of the *L. paracasei* strains (*n* = 75) based on 243 conserved single-copy marker genes. Our analysis revealed that *L. paracasei* Lbs2 (reassembled genome), *L. paracasei* subsp. *tolerans* DSM20258, and *L. paracasei* Lpc-37 (higher degree of genome completeness) strains are originated from a common node, but only the former two share close proximities (Figure 5).

For this reason, strain DSM20258 is employed for further analysis along with the reassembled genome (*L. paracasei* Lbs2). Interestingly, hierarchical clustering of COG categories frequency revealed that both the Lbs2 and DSM20258 strains are in the same cluster (Figure 2).

### 3.4. Horizontal Gene Transfer Analysis Indicates L. paracasei Lbs2 Strain Acquired Important Niche-Specific Genes

Horizontal gene transfer (HGT) can be interpreted as the gaining of genetic material from other organisms without being its offspring. This event is an important force for the bacterial genome evolution. In the Lbs2 strain, we identified five genomic islands (GIs) using the IslandViewer4 tool (Appendix A). These GIs hold a total of 205 (193 without duplicate) genes and most of them are putative (Appendix A). The lengths of these GIs range from 4671 to 163,188 bp. Interestingly, thiamin biosynthetic genes (*thiE_2, thiM_2*) are found in the largest GI (GI 4), which are also unique to the Lbs2 strain. GI 2 and GI 5 constituted only hypothetical genes, whereas GI 1 contained important genes like *inlJ_1*, *dps*, and GI 3 with *recX* gene (Appendix A). Further, we carried out a blast of horizontally transferred genes against the AMR (antimicrobial resistance) database [51] and found that none of the genes codes for an antimicrobial resistance gene. But blasting of the remaining genes (*n* = 2187) against the AMR database indicates that only three genes (e.g., two beta sub-unit of RNA polymerase and one Elongating factor Tu) are potentially antimicrobial resistant (Appendix A). This indicates that the antimicrobial resistances genes present in Lbs2 are not acquired by horizontal transfer.

The pan-genome of *L. paracasei* strains (*n* = 75) contains several multidrug resistance (MDR) and antimicrobial resistance (AMR) genes (Appendix A), which may be intrinsic or acquired via a horizontal gene transfer mechanism. Intrinsic resistance is a chromosomally encoded inherent feature in bacteria and is not movable. The best example of intrinsic resistance in Lactobacilli is resistance to vancomycin [52]. Vancomycin blocks bacterial growth by affecting peptidoglycan synthesis, which is an essential component for the cell wall of gram-positive bacteria. Interestingly, pathway analysis throughout *L. paracasei* strains (*n* = 75) revealed the abundance of vancomycin resistance genes (Appendix A). Also, a study on *L. rhamnosus* suggests that the bacteria with vancomycin resistance did not transfer vancomycin resistance genes to recipients [52]. Very early reports on the presence of vancomycin resistance genes in *Lactobacillus rhamnosus* suggests that the genes have diverged completely from the vancomycin resistance genes of Enteroccocal strains [53]. There are already a large number of studies pointing towards the intrinsic nature of vancomycin resistance to *Lactobacillus* species. Generally, this antibiotic binds to the d-alanine residue of the peptidoglycan residue and blocks the peptidoglycan biosynthesis. In the case of Lactobacilli, the mechanism of resistance involves the substitution of the *D*-alanine residue with either a *D*-lactate or *D*-serine, preventing the binding of the antibiotic [54]. Vancomycin is generally used to treat a number of gram positive bacterial infections in the gut. Resistance to vancomycin may have been an ancient event and may have originated in Lactobacilli itself as a parallel event rather than an acquired event from the environment.

On the contrary, acquired resistance due to horizontal gene transfer poses a threat to nonpathogenic bacteria. The exchange of virulence genes from commensals to Lactobacilli and resistance genes from Lactobacilli to intestinal commensals inside the colon can totally change the genotypic profile of commensals and Lactobacilli [52].

### 3.5. Extracellular Properties of the L. paracasei Strains

In Lactobacilli, the LPXTG motif-containing extracellular proteins are anchored to the cell wall by sortase enzymes [55]. These proteins play important roles in adhesion and colonization. Several experiments have suggested that the act of sortases and their substrates is vital for deciphering various probiotic modes of action [6]. In the pan-genome (*n* = 75) of *L. paracasei*, we identified a total of 415 LPXTG motif-containing proteins and 141 sortase enzymes (Appendix A). The number of the LPXTG motif and sortase enzymes varies greatly among the *L. paracasei* strains (0–12 for LPXTG and 0–4 for sortase). The highest numbers of LPXTG proteins (*n* = 12) are found in the 525_LPAR strain. Interestingly, both CNCMI-4648 (genome completeness = 92.45% as per CheckM analysis) and Lpp70 (genome completeness = 99.27% as per CheckM analysis) strains lacking LPXTG motif-containing proteins also lacked sortase protein, indicating a possible gene loss event. Additionally, LPXTG motifs and sortase enzymes are also identified in strains Lpc-37 (9 vs. 3), Lbs2 (6 vs. 2), and DSM20258 (2 vs. 1), respectively.

Identification of these proteins in Lactobacilli provides information about their key roles in resolving nutrient uptake through proteinase P and with positive probiotic traits, such as mucus barrier function, adhesion, and immune signaling [56]. Moreover, sortase expression signals in Lactobacilli have been utilized to develop gastrointestinal tract targeted oral vaccines [56].

### 3.6. Various Stress Factors Characterized Across the L. paracasei Strains

Lactobacilli have faced several environmental stress factors, such as low pH, bile salts, and oxidative and osmotic stress, during their transit through the diverse habitat. Our analysis has revealed that the number of genes associated with oxidative stress, osmotic stress, and salt stress varies greatly across the *L. paracasei* strains (*n* = 75), while the rest of the stress-related gene numbers remains fairly constant irrespective of their environment (Appendix A). In this study, we have also found the highest number of genes linked with oxidative stress (*n* = 11,416) compared to other stress linked genes, such as osmotic (*n* = 6359), salt (*n* = 3721), nitrosative (*n* = 617), heat (*n* = 155), DNA damage (*n* = 773), cold (*n* = 0), acid (*n* = 0), and bile (*n* = 76) stresses across the all strains of *L. paracasei*. A total of 330 stress linked genes are identified in the Lpc-37 strain alone, whereas the Lbs2 and DSM20258 strain contained 260 and 257 stress associated genes each. These genes can change the activity of the general metabolism, membrane components, and transporter systems of the cell in a hostile environment [57]. Glycine betaine/proline transport system (ATP-binding protein) and glycine betaine/carnitine transport (ATP-binding protein) encoding genes are found to be predominant among the osmotic stress related genes, which are thought to be involved in the accumulation of glycine, betaine, and carnitine in response to increased external osmolarity [58]. Interestingly, the predominant oxidative stress encoding genes are annotated as calcium transporting ATPase followed by a hydrogen peroxide-inducible gene activator. It is proposed that the hydrogen peroxide-inducible genes activator has a crucial role in hydrogen peroxide scavenging for the repair of oxidative damage [59]. In addition, an important enzyme, bile salt hydrolase related to adaptation to the gut environment is identified among the *L. paracasei* strains except for the Lbs2 strain that is isolated from the gut (Appendix A). Notably, the Lbs2 strain was cultivated for many generations outside of its native environment and may cause the loss of the gene encoding the bile salt hydrolase (*bsh*) enzyme. Apart from that, as the Lbs2 draft assembly covered 91.28% of the genome, this could also be the reason for not capturing the *bsh* gene. Importantly, the presence of this enzyme in plant isolates such as Lpl14, Lpp189, Lpp46, Lpp49, and CNCMI-2877 indicates that these strains can be good probiotic candidates (Appendix A). Moreover, the heatmap of stress linked genes has shown a closer relationship between Lbs2 and DSM20258 strains, as they are part of the same cluster (Appendix A).

### 3.7. Bacteriocins Identified Among the Lactobacillus paracasei Strains

Bacteriocins are small cationic peptides with a key function like quorum sensing. In the genomes of *L. paracasei* strains (*n* = 75), a total of 191 bacteriocins are predicted with BAGEL4 (Appendix A). The highest numbers of bacteriocins are found to be present in 275_ LPAR, FAM 18149, Lpp 122, and Lpp 219 strains, but absent in DSM20258, Lpp48, and Lpp70 strains. Moreover, a lone copy of bacteriocin is found in Lbs2, Lpc-37, CAUH 35, Lpp 189, Lpp 228, Lpp 49, and TMW1.1434 strains (Appendix A). Based on their structural properties, bacteriocins from Lactobacilli are divided into three major classes. Class I bacteriocins, the lantibiotics, are small peptides that undergo extensive posttranslational modifications. Class II bacteriocins are unmodified, heat-stable peptides. Class III bacteriocins are the least characterized to date [60]. Bacteriocins of the *L. paracasei* Lbs2 strain, identified as Enterolysin_A, belongs to Class III types of bacteriocins (Appendix A). The sequence analysis of Enterolysin A suggested that this bacteriocin consists of two separate domains, an *N*-terminal catalytic domain and a C-terminal substrate recognition domain [58]. This Enterolysin_A played a crucial role in cell-wall degradation of pathogens [60]. Among other strains of Lactobacilli, the class II bacteriocins are pretty common. Examples of this class of bacteriocins predicted in other members of Lactobacilli are LSEI_2386, LSEI_2163, Thermophilin_A, and Carnocin_CP52 (Appendix A). Already, there are several reports on the structural characterization of different bacteriocins [61]. It has also been reported that Class II bacteriocins might contain peptides with a double-glycine leader and hypothesized that the existence of a disulfide bridge among these peptides plays an important role in antibacterial activity [62]. In recent times, the majority of studies have focused on bacteriocins producing probiotics, which can inhibit the growth of gut pathogens. It is thought that the production of bacteriocins could provide probiotic functionality in three different ways. Initially, it may act as colonizing peptides, building the dominance of a producer into an already occupied habitat [63]. Secondly, bacteriocin may play the role of killing peptides, where it is directly inhibiting competing strains or pathogens [64]. Finally, bacteriocins may serve as signaling peptides, either signaling other bacteria via quorum sensing and signaling cells of the host immune system [65]. It has also been reported that bacteriocins can be deployed as anticancer agents, either through their impact on cancerous cells or the suppression of bacteria associated with the initiation of disease [66].

### 3.8. A Wide Distribution of Mobile-Genetic Elements and CRISPR-Cas Systems

Using PHASTER, a total of 276 prophages were detected among the 75 *L. paracasei* strains. The maximum number of prophage regions (*n* = 10) were observed in the strain EG9, whereas only one prophage region was detected in 13 strains (275_LPAR, B3, CNCMI-2877, CNCMI-4648, Lpp46, Lpp49, Lpp70, Lpp123, Lpp126, Lpp189, Lpp221, Lpp227, Lpp228, and Lbs2). In Lbs2, the observed prophage region is highly related to *Lactobacillus* phages and encodes seven proteins, but was found to be incomplete (5.4 Kb). In the strain DSM20258, two incomplete prophage regions of length 16.4 and 18.6 Kb, respectively, coding 10 proteins each, were observed, which are also highly related to *Lactobacillus* phages. Moreover, strain Lpc-37 has an incomplete prophage (14 Kb) and two putative prophages (25.8 and 41.6 Kb) regions (Appendix A). It is believed that the presence of prophages in Lactobacilli genomes may protect them from superinfection by other phages or plasmid [67].

Prokaryotic genomes possess a large number of CRISPR loci, which play a vital role in controlling horizontal gene transfer [68]. It is a well-known fact that some bacteria have gained the CRISPR-cas system as a defense system against phage invasion. A total of 238 CRISPR loci across the *L. paracasei* genomes (*n* = 75) were predicted using a stand-alone tool, CRISPR finder (Appendix A). CNCMI-2877, Lpp226, Lpp229, and Lpp41 strains are lacking CRISPR loci, evidently because these genomes are incomplete. The DmW_181 strain carries the maximum number of CRISPR loci despite having a very fragmented assembly with 127 scaffolds. In addition, the number of CRISPR loci predicted in Lbs2, DSM20258, and Lpc-37 strains are two, one, and three, respectively. In Appendix A, CRISPR loci of *L. paracasei* Lbs2 are depicted.

We have also identified IS elements using the ISsaga platform, which may contribute to bacterial genome evolution. From *L. paracasei* Lbs2, *L. paracasei* subsp. *tolerans* DSM20258, and *L. paracasei* Lpc-37 genomes, a total of 10, 5, and 98 IS elements are predicted (Appendix A). Compared to the 74 genomes of *L. paracasei,* FAM18149 exhibited a higher number of IS elements that may suggest a higher potential for genome plasticity.

### 3.9. Broad Range Carbohydrate-Active Enzymes (CAZymes) and Carbohydrate Transporters Identified in the Pan-Genome of Lactobacillus paracasei

Carbohydrates are the prime source of energy for all organisms. A major fraction of genes found in the pan-genome (*n* = 75) of *L. paracasei* are associated with carbohydrate metabolism and transport; thus, we compared CAZymes encoding genes across the strains. Currently, a total of 153 GH (glycoside hydrolases), 106 GT (glycosyl transferase), 27 PL (polysaccharide lyase), 83 CBM (Carbohydrate-Binding Module), 16 CE (carbohydrate esterase), 15 AA (auxiliary activities) families, and 42 GH13, 37 GH43 subfamilies are present in the CAZy database (http://www.cazy.org).

In this study, 32 GHs, 17 GTs, 2 PLs, 7 CBMs, 7 CEs, 4 AAs families, and 7 GH13, 1 GH43 subfamilies are identified in the *L. paracasei* strains (Appendix A). Our analysis also revealed that the *L. paracasei* strains carbohydrate-activating enzyme numbers ranged from 72 (present in the Lpp123 strain) to 115 (present in the Lpc-37 and NRIC0644 strains). Among the 115 enzymes of the Lpc-37 strain, 46 are identified as GHs, 35 as GTs, 16 as CEs, 11 as CBMs, 5 as AAs, and 2 as PLs. Compared to the 115 enzymes of the *L. paracasei* Lpc-37 strain, the reassembled genome (*L. paracasei* Lbs2) and its closest one (*L. paracasei* subsp. *tolerans* DSM20258) contained 86 (37 are identified as GHs, 22 as GTs, 14 as CEs, 6 as CBMs, 5 as AAs, and 2 as PLs.) and 73 (24 are identified as GHs, 24 as GTs, 15 as CEs, 6 as CBMs, 4 as AAs, and 0 as PLs.) enzymes, respectively. To transform undigested carbohydrates, present in the gastrointestinal tract or in the environment, several glycosyl hydrolase family (GH) enzymes are being used by Lactobacilli. In *L. paracasei* Lbs2 β-xylosidase and α-*L*-iduronidase (GH39), enzymes are unique with respect to Lpc-37 and DSM20258 strains, indicating that the Lbs2 strain probably needs these enzymes in its ecological niche.

Interestingly, cellulose synthase (GT2), a key enzyme for cellulose biosynthesis, is found to be predominant across the 75 strains of *L. paracasei*, which could hoard cellulose on the cell wall surface as an extracellular matrix for cell adhesion and biofilm formation to defend itself from the surrounding environment [69,70,71]. Noticeably, Lactobacilli uses glycogen as carbohydrate storage forms, and it has been reported that they are synthesizing glycogen to engage with more diverse habitats [72]. Glycogen synthase (GT5) and glycogen phosphorylase (GT35) enzymes are involved in glycogen synthesis. The current study showed that the GT5 and GT35 families encoding gene numbers remain the same among the Lpc-37, Lbs2, and DSM20258 strains and could probably be required for adaptation in diverse habitats.

Furthermore, potential carbohydrate transporters are identified in *L. paracasei* strains using the GO database, which ranged from 99 (present in the Lpp126 strain) to 149 (present in the IIA strain). *L. paracasei* Lbs2, *L. paracasei* Lpc-37, and *L. paracasei* subsp. *tolerans* DSM20258 contain 114, 106, and 142 potential carbohydrate transporters, respectively (Appendix A).

### 3.10. In-Depth Comparative Analysis of L. paracasei Lbs2 Against L. paracasei subsp. tolerans DSM20258 and L. paracasei Lpc-37

To further examine the technological and probiotic properties of *L. paracasei* Lbs2, we carried out a comparative genomic analysis against *L. paracasei* subsp. *tolerans* DSM20258, and *L. paracasei* Lpc-37. *L. paracasei* Lbs2 isolated from a healthy human gut (north Indian) has a genome size of 2.50 Mb. The genome size of *L. paracasei* subsp. *tolerans* DSM20258 and *L. paracasei* Lpc-37 are 2.36 Mb and 3.16 Mb, respectively. The detail of genomic features among these three strains are elaborated in Table 2.

Our analysis using OrthoVenn also unveiled that the number of core genes among these three strains is 1678 (Appendix A), higher than that computed for the pan-genome of *L. paracasei* (Figure 3A). The distribution of the core COG functional categories are depicted in Appendix A; the majority of them are associated with translation, ribosomal structure and biogenesis (J), amino acid transport and metabolism (E), carbohydrate transport and metabolism (G), and function unknown (S) (Appendix A). Additionally, singletons of the three strains allocated in all COG categories, most of them affiliated to carbohydrate transport and metabolism (G), mobilome: prophages and transposase (X) and defense mechanism (V) (Appendix A). It could be suggested that the high prevalence of proteins in the G (Carbohydrate transport and Metabolism) COG category have a direct effect on the niche diversity in which organisms can grow. On the contrary, mobilome: prophages and transposase can create genome plasticity.

Furthermore, whole genome alignments based on progressive MAUVE among the three strains lack of extensive synteny (Appendix A) corroborates with the absence of thiamin biosynthetic gene cluster in the DSM20258 strain (Appendix A). However, the glycin/betain transporter gene cluster has shown perfect synteny among these strains (Appendix A).

### 3.11. Evaluating the Technological and Probiotic Traits of L. paracasei Lbs2

Probiotic properties of *L. paracasei* subsp. *tolerans* DSM20258 have been extensively studied in the past [1]. The genome of the DSM20258 strain contains important probiotic traits, like bile salt hydrolase. Surprisingly, this trait was absent from the gut isolate *L. paracasei* Lbs2, which may be due to genome incompleteness. In addition, gene encoding bacteriocin were not present in the DSM20258 genome, while the Lbs2 genome contained enterolysin_A like bacteriocin. Furthermore, sortase-dependent cell surface proteins in the DSM 20258 strain, which may interact with the host, are also found to be present in the Lbs2 strain.

We then examined the technological aspects of gut isolate *L. paracasei* Lbs2, keeping in mind that a human being cannot synthesize most of the vitamins and for this it needs to be outsourced. Probiotic bacteria, located in the human gut, such as Lactobacilli, can *de novo* synthesize and provide vitamins to the human body. Gut microbiota in humans, capable of synthesizing vitamin K, and most of the water soluble B vitamins, such as pyridoxine, folates, riboflavin, cobalamin, and thiamin. Among the water soluble B vitamins, thiamin (vitamin B1) as thiamine pyrophosphate (TPP), played a crucial role in host energy metabolism since it acts as a co-factor for major metabolic pathways, such as the pentose phosphate pathway, glycolysis, and Kreb’s cycle (Figure 6A). The pentose phosphate pathway is needed for steroids, nucleic acids, fatty acids, and the aromatic amino acid biosynthesis. These products from the pentose phosphate pathway are used as precursors of different neurotransmitters and other bioactive compounds vital for brain function [73]. Interestingly, the thiamin biosynthetic gene cluster along with the TPP riboswitch is found to be present in *L. paracasei* Lbs2 and *L. paracasei* Lpc-37 genomes but is absent in DSM20258 (Appendix A). In this study, we have shown the plausible role of these genes in thiamin biosynthesis (Figure 6B).

Surprisingly, from the pan-genome analysis, it was found that two thiamin biosynthesis related genes (*thiE_2* and *thiM_2*) are unique to the *L. paracasei* Lbs2 genome (Appendix A). Hence, the presence of unique thiamin biosynthesis genes in the *L. paracasei* Lbs2 genome could be one of the reasons for probiotic adaptation leading to improved host energy metabolism.

## 4. Conclusions

In this study, we reassembled and analyzed gut isolated Lbs2 strains earlier identified as *L. casei* species and placed it under the *L. paracasei* species, after careful genomic analysis. Our analysis was based on pair wise genome distance (ANI and AAI) calculations, which also indicated that many strains of *L. casei* were classified wrongly in the NCBI genome repository, and this needed to be reclassified as *L. paracasei*.

Reclassification followed by the pan-genome analysis of *L. paracasei* strains revealed that the Lbs2 strain holds a small core-proteome; most of them are ribosomal proteins. Furthermore, phylogeny based on core-genome indicates that the *L. paracasei* subsp. *tolerans* DSM20258 strain is more closely related to *L. paracasei* Lbs2 as compared to other strains. Interestingly, probiotic features of the DSM20258 strain are established in earlier studies [1], which make it easier for us to find out the probiotic traits, such as sortase-dependent cell surface proteins, bacteriocins, etc., present in the Lbs2 strain. However, important probiotic traits, like bile salt hydrolase, are missing from the Lbs2 strain compared to DSM20258; this may be due to genome gaps. Finally, we have also identified a thiamin biosynthetic gene cluster in the Lbs2 strain that is thought to be involved in enhancing host energy metabolism. Surprisingly, *thiE_2* and *thiM_2* genes are found to be unique in the Lbs2 strain. These unique genes are also located in the genome islands on the Lbs2 strain.

In summary, our results indicate further study is required with other members of the *L. casei* group, like *L. rhamnosus* and *L. casei,* for potential reclassification among closely related members of the *L. casei* group and to better understand the more technical aspects lying behind the host adaptation.

## Figures and Tables

**Figure 1 microorganisms-07-00487-f001:**
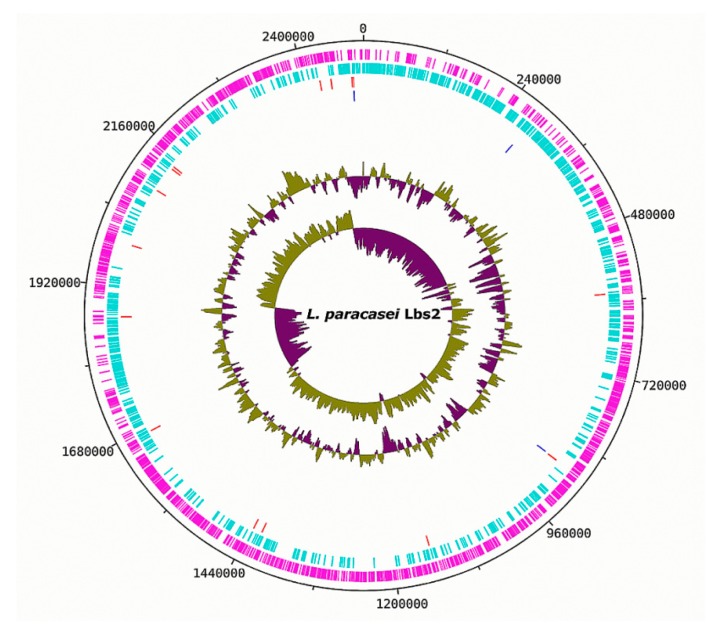
Circular map of the *L. paracasei* Lbs2 genome. Labeling from outside to inside of the circle, each ring carries information of the genome: coding sequences (CDSs) on the forward strand (magenta); CDSs on the reverse strand (cyan); tRNA genes (red); rRNA genes (blue); GC content; GC skew.

**Figure 2 microorganisms-07-00487-f002:**
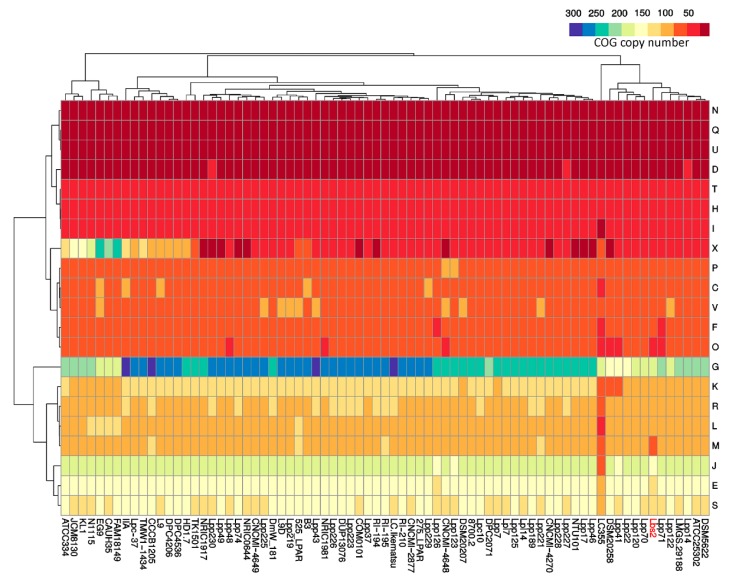
Cluster of Orthologous Groups (COG) frequency heatmap based on hierarchical clustering. The horizontal axis depicts functional COG categories, and the vertical axis represents 75 *L. paracasei* strains. Genome of interest ‘Lbs2’ is marked as red.

**Figure 3 microorganisms-07-00487-f003:**
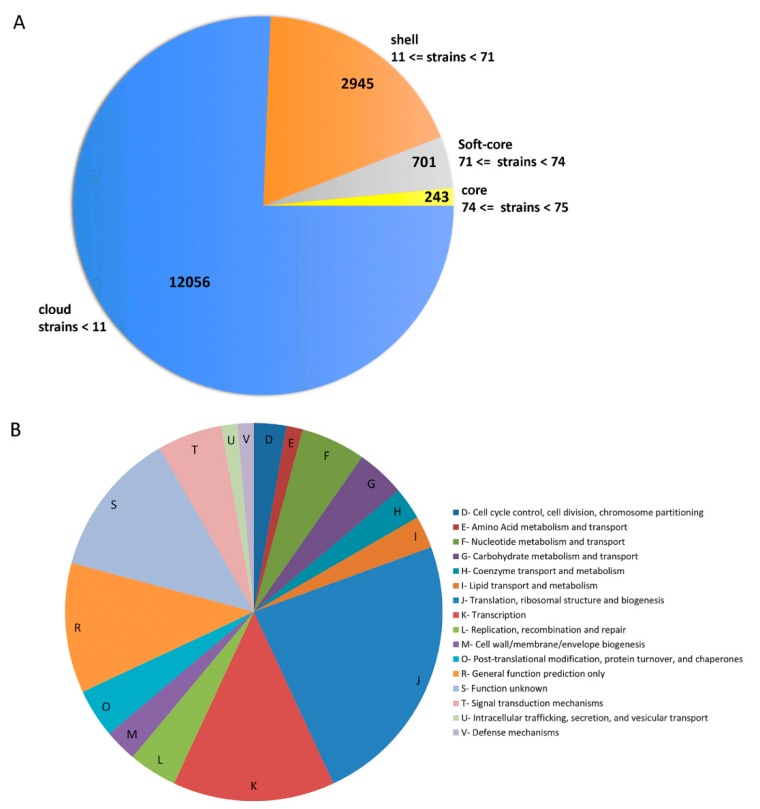
Pan-genome analysis of 75 *L. paracasei* strains. (**A**) Pie-chart representing core and accessory genes distribution. (**B**) Pie-chart representing the distribution of COG categories of 243 core-gene families.

**Figure 4 microorganisms-07-00487-f004:**
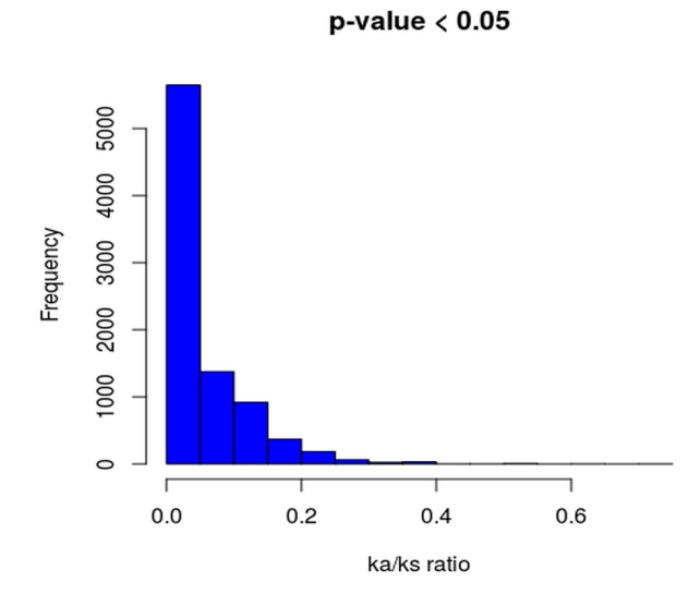
Histogram illustrating Ka/Ks ratios of each core gene.

**Figure 5 microorganisms-07-00487-f005:**
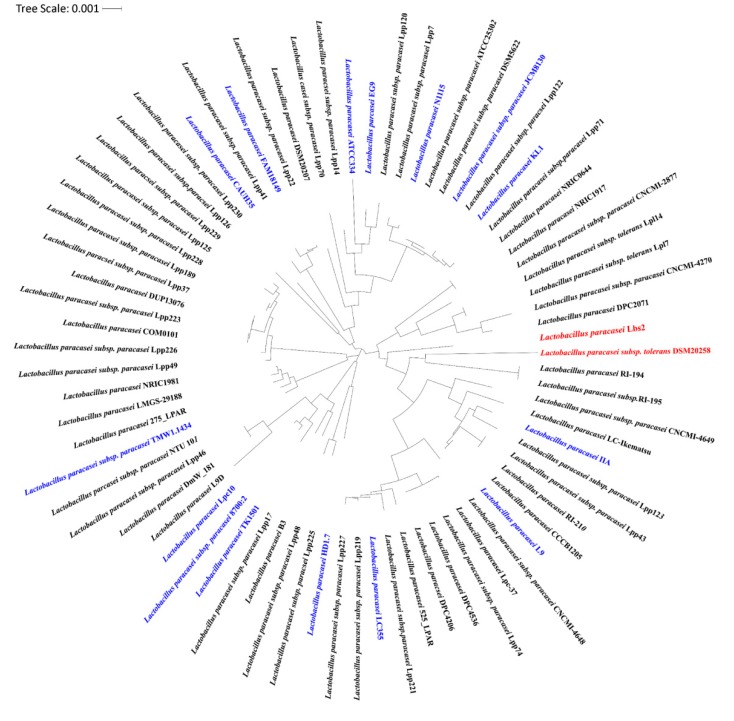
Whole genome phylogenetic tree of *L. paracasei* strains was inferred using the maximum likelihood method. The tree was built on the basis of the core-genome and it is presented as a cladogram. Complete genomes are marked as blue, while the genome of interest and its closest one are marked as red.

**Figure 6 microorganisms-07-00487-f006:**
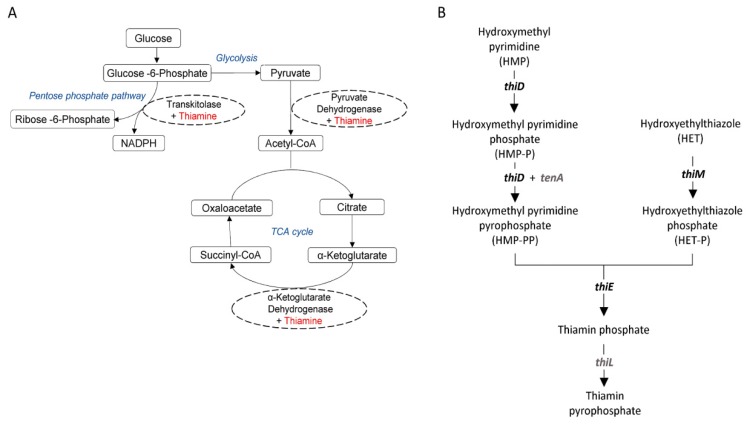
The schematic illustration of pathways. (**A**) The role of thiamin (marked as red) as a co-factor in major metabolic pathways (marked as blue). (**B**) Genes involved in thiamin biosynthesis.

**Table 1 microorganisms-07-00487-t001:** Proposed name of *L. casei* strains based on the Average Nucleotide Identity (ANI) and Average Amino Acid Identity (AAI) calculation.

Existing Name in NCBI Database (GenBank)	Proposed Name
*Lactobacillus casei* 12A (Complete genome)	*Lactobacillus paracasei* 12A
*Lactobacillus casei* 12A (Draft)	*Lactobacillus paracasei* 12A
*Lactobacillus casei* 1316.rep1_LPAR (Scaf no.170)	*Lactobacillus paracasei* 1316.rep1_LPAR
*Lactobacillus casei* 1316.rep2_LPAR (Scaf no.264)	*Lactobacillus paracasei* 1316.rep2_LPAR
*Lactobacillus casei* 21/1	*Lactobacillus paracasei* 21/1
*Lactobacillus casei* 32G	*Lactobacillus paracasei* 32G
*Lactobacillus casei* 5b	*Lactobacillus paracasei* 5b
*Lactobacillus casei* 844_LCAS	*Lactobacillus paracasei* 844_LCAS
*Lactobacillus casei* A2-362 (Scaffold no. 162)	*Lactobacillus paracasei* A2-362
*Lactobacillus casei* A2-362 (Contig no. 167)	*Lactobacillus paracasei* A2-362
*Lactobacillus casei* BD II	*Lactobacillus paracasei* BD II
*Lactobacillus casei* BL23	*Lactobacillus paracasei* BL23
*Lactobacillus casei* BM-LC14617	*Lactobacillus paracasei* BM-LC14617
*Lactobacillus casei* CRF28	*Lactobacillus paracasei* CRF28
*Lactobacillus casei* DPC6800	*Lactobacillus paracasei* DPC6800
*Lactobacillus casei* DSM 20011	*Lactobacillus paracasei* DSM 20011
*Lactobacillus casei* GCRL163	*Lactobacillus paracasei* GCRL163
*Lactobacillus casei* HDS-01	*Lactobacillus paracasei* HDS-01
*Lactobacillus casei* HZ-1	*Lactobacillus paracasei* HZ-1
*Lactobacillus casei* KL1-Liu	*Lactobacillus paracasei* KL1-Liu
*Lactobacillus casei* LC2W	*Lactobacillus paracasei* LC2W
*Lactobacillus casei* LOCK919	*Lactobacillus paracasei* LOCK919
*Lactobacillus casei* Lc-10	*Lactobacillus paracasei* Lc-10
*Lactobacillus casei* Lc1542	*Lactobacillus paracasei* Lc1542
*Lactobacillus casei* LcY	*Lactobacillus paracasei* LcY
*Lactobacillus casei* Lpc-37 (Contig no.150)	*Lactobacillus paracasei* Lpc-37
*Lactobacillus casei* M36	*Lactobacillus paracasei* M36
*Lactobacillus casei* MJA12	*Lactobacillus paracasei* MJA12
*Lactobacillus casei* T71499	*Lactobacillus paracasei* T71499
*Lactobacillus casei* UCD174	*Lactobacillus paracasei* UCD174
*Lactobacillus casei* UW1	*Lactobacillus paracasei* UW1
*Lactobacillus casei* UW4 (Contig no. 122)	*Lactobacillus paracasei* UW4
*Lactobacillus casei* UW4 (Contig no. 144)	*Lactobacillus paracasei* UW4
*Lactobacillus casei* W14	*Lactobacillus paracasei* W14
*Lactobacillus casei* W16	*Lactobacillus paracasei* W16
*Lactobacillus casei* W56	*Lactobacillus paracasei* W56
*Lactobacillus casei* Z11	*Lactobacillus paracasei* Z11
*Lactobacillus casei* Zhang	*Lactobacillus paracasei* Zhang

**Table 2 microorganisms-07-00487-t002:** Genome assembly statistics of the three *Lactobacillus paracasei* strains (Lpc-37, Lbs2, DSM20258).

Strain			
Features	Lpc-37	Lbs2	DSM20258
Source	Microbial food product	Human Gut	Not available
Genome Status	Draft	Draft	Draft
Accession Number	NOKL00000000.1	JPKN00000000.3	AYYJ00000000.1
N50 (bp)	3,112,081	10,992	14,516
L50	1	68	49
Completeness (%)	100	91.28	97.19
Contamination (%)	0	0.87	0
Size (Mb)	3.16	2.50	2.36
GC%	46.33	46.97	46.44
Genes	3125	2380	2424
Proteins	3010	2308	2339
t-RNA	59	20	37
r-RNA	15	3	2
Other-RNA	41	49	46

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
