# Peer review of "Reanalysis of Lactobacillus paracasei Lbs2 Strain and Large-Scale Comparative Genomics Places Many Strains into Their Correct Taxonomic Position"

_microorganisms, 2019, doi:10.3390/microorganisms7110487_

Round 1
Reviewer 1 Report
This work presents a nice annotation of bacteria belonging to Lactobacillus paracasei which are broadly of interest because they are present in food products and can serve as probiotics. This is increasingly important as we learn more about the multitude of roles that the gut microbiotome can play in human health. Having a complete and detailed annotation of these bacteria is a service to the scientific community, although it certainly is appropriate for a specialized journal. The reasoning for the choice of genes was sound, and it was a sound point that typical 16S rRNA is not sufficient to resolve at the strain level. I also appreciated the transparency of including all the numerical data used to make heat maps in the supplemental. That being said, I have a few minor comments:
On page 7, lines 14-19, the authors mention that intrinsic resistance in Lactobacilli to vancomycin is not transferred through horizontal gene transfer. Can the authors expound upon the significance of this and what the evolutionary reason for it might be? For figure S7 and the discussion of the bacteriocin BCGs present, can the authors speculate on their structure or perhaps show some predicted structures? It might further illustrate their role in the bacteria. Figure S13 is largely unclear and difficult for the reader to extract any informative content In Figure S1 and several others in the supplemental, red green heat map shading is challenging for colorblind individuals. The community has moved more towards blue-yellow heat maps so that this is not a barrier to understanding the scientific content.
Author Response
To
The Editor in Chief
MDPI-Microorganisms,
We thank the anonymous reviewers. Their comments were highly valuable for providing the best relevant information to the journal readers. We believe this improved the quality of the manuscript immensely. Here is a point to point response to the reviewer’s comments.
We will be happy to answer any other query/concern the reviewers or the editors may have.
With Best Regards
Sincerely
Sucheta Tripathy
Reviewer 1 Comments and response
Q1. On page 7, lines 14-19, the authors mention that intrinsic resistance in Lactobacilli to vancomycin is not transferred through horizontal gene transfer. Can the authors expound upon the significance of this and what the evolutionary reason for it might be?
Our Response: The significance and plausible evolutionary reason for intrinsic resistance in Lactobacilli to vancomycin with respect to horizontal gene transfer is discussed in page 6-line number 30-39 [Highlighted in yellow]. In fact, it is a well-known fact that Vancomycin resistance in Lactobacilli is intrinsic and may have an ancient acquired mutation. We have provided additional references for that in that section.
Q2. For figure S7 and the discussion of the bacteriocin BCGs present, can the authors speculate on their structure or perhaps show some predicted structures? It might further illustrate their role in the bacteria.
Our Response:Thank you for pointing this out. There are 3 classes of Bactericins and the peptide sequences of these bacteriocins are quite divergent. Now we have updated Supplementary File S10 with Bacteriocin sequences from all the strains. The detailed structure prediction and illustrating their role may be presently out of scope. We have added additional explanations for bacteriocins and the roles and their importance in the main manuscript in pages 8 in line 7 – 19 in yellow.
Q3. Figure S13 is largely unclear and difficult for the reader to extract any informative content
Our Response: Figure S13 was generated using Muave and we concatenated all of the scaffolds. The rationale behind putting this image is to show that there are no major regions of the genomes missing. Although it is unclear on which block aligns with which block, it shows a global picture. If you suggest removal of this figure, we can gladly oblige. For now, we have updated the figure legends.
Q4. In Figure S1 and several others in the supplemental, red green heat map shading is challenging for colorblind individuals. The community has moved more towards blue-yellow heat maps so that this is not a barrier to understanding the scientific content.
Our Response:Thanks for pointing this out. We have now updated the colors so that it is more soothing for color blind individuals.
Reviewer 2 Report
Summary
The authors compared the genome of Lactobacillus paracase and Lactobacillus casei, reclassified some strains into correct species. Additionally, it reassembled the genome of L. paracase Lbs2 with better completeness. This manuscript also clearly profiles the genome and identified the important genes of Lactobacillus paracasei Lbs2 and other L. paracase strains. This can help us to compare the different traits between L. paracase Lbs2 and other strains. However, the figures are missing (not attached?) so I can not understand the result only by the words.
Major:
Absent of Figures. I can not see the results. In S3 table, it’s better to sort the lists by the completeness, so the readers can compare if the results of CheckM and BUSCO are similar. In Figure S6, the color of heatmap and the legend seem different. Besides, why the colors of cold(gene number=0) and acid(n=0) are same with bile(n=76), heat(n=115), dna damage(n=773) and nitrosative(n=617)?
Minor:
Lack of unit with N50. In S4 table, what does the blue color mean? Please note. There are blank sheets in Table S8. Please remove. The supplementary figures are not in order. (S4 -> S6 ->S5) Page 8 lane 50, Figure8? Figure S8?
Author Response
To
The Editor in Chief
MDPI-Microorganisms,
We thank the anonymous reviewers. Their comments were highly valuable for providing the best relevant information to the journal readers. We believe this improved the quality of the manuscript immensely. Here is a point to point response to the reviewer’s comments.
We will be happy to answer any other query/concern the reviewers or the editors may have.
With Best Regards
Sincerely
Sucheta Tripathy
Reviewer 2 Comments
Major:
Q1. Absent of Figures. I can not see the results.
Our Response: Our sincere apologies for the figure disappearance. We did upload the figures, may be due to some technical difficulty they disappeared. We are uploading this again. Please let us know if they are visible.
Q2. In S3 table, it’s better to sort the lists by the completeness, so the readers can compare if the results of CheckM and BUSCO are similar.
Our Response: Many Thanks for pointing this out. This is indeed a very valued suggestion. We have sorted the table as desired by the reviewers.
Q3. In Figure S6, the color of heatmap and the legend seem different. Besides, why the colors of cold(gene number=0) and acid(n=0) are same with bile(n=76), heat(n=115), dna damage(n=773) and nitrosative(n=617)?
Our Response: Now it has been corrected. The legend of Figure S6 is marked in yellow. In heat map (Figure S6), plotting was made on the basis of the gene numbers of individual strains (not on the basis of total genes of 75 strains). The legend is self-explanatory.
Minor:
Q1. Lack of unit with N50.
Our Response: We added the unit of N50 and that is in bp in page no. 5 and line number 8.
Q2. In S4 table, what does the blue color mean?
Our Response: In Table S4, the blue color indicates the strains are of plant origin. Now the excel file itself is annotated with this for reducing confusion.
Q3. Please note. There are blank sheets in Table S8. Please remove.
Our Response: In Table S8 all the blank sheets have now been removed.
Q4. The supplementary figures are not in order. (S4 -> S6 ->S5)
Our Response: We have now corrected the order of the figures, now it is [S4->S5->S6].
Q5. Page 8 lane 50, Figure8? Figure S8?
Our Response: We have now renamed Figure 8 to Figure S8A in line 43 and Figure S8B in line 47.
Round 2
Reviewer 2 Report
Summary
This paper compared the genome of Lactobacillus paracase and Lactobacillus casei, reclassified some strains into correct species. Additionally, it reassembled the genome of L. paracase Lbs2 with better completeness. This paper also clearly profiles the genome and identified the important genes of Lactobacillus paracasei Lbs2 and other L. paracase strains. This can help us to compare the different trairs between L. paracase Lbs2 and other strains. However, use too many tables to tell the result. The results should be well organized and find a good figure to explain them. In addition, some figures are not appropriate and can’t well and easily explain a concept.
Major:
Use too many tables to explain the result, it seems that you didn't well organize them and just give all your raw data. Please try to organize them and use appropriate figures to show your results. Following up major 2 of last time. In S3 table, it’s better to sort the lists by the column of completeness (CheckM) and completeness BUSCOs (BUSCO), so the readers can compare if the “ranking” of CheckM and BUSCO are similar. In addition, what is the reason to use two methods to evaluate the degree of completeness? Maybe it is enough using only one method? Within all the heatmaps, some of the rows are the same color but belong to different cluster. Suggest to use a smooth gradient color (not only 9 colors) and change the scale and breaks of the color key, so that the readers can easily recognize the difference between each rows and columns. In Figure 3B, heatmap is not the appropriate figure, as pie chart is better if you only want to explain which function is more. The core COG copy number should be same in different strains, so there is no need to do heatmap and cluster. In Figure 4, it only needs one figure to explain Ka/Ks ratio. In Figure S11B, if you use the data of core genes to draw heatmap, the numbers should be same in these 3 strains. Why they are different color in Lpc37? According to your discussion (page 10 lane 4-16), pie chart may be more appropriate if you only want to explain which function is more. In Figure S12 and Table S15, why most of COG categories affiliated to M, G, V? Not G, X, V (the total number is higher)? Is it a work simply to correct the authors' previous work (bad assembly leading to mis-identification of strains)? If so, I am not sure this work (erratum) justifies a new research paper? P1 - The authors claim that in NCBI, strains of L. casei species are misclassified and these strains are needed to be placed under L. paracasei species. I am not sure how solid the evidences are there to call "misclassified" and what are the alternative evidences that NCBI established at the first place. The authors should present clear and fair arguments before they claim so called "misclassified". This is particularly important as it is the foundation to the whole manuscript. P2 top 2 paragraphs -- Not sure how those features are related to the comparative genomics between L. paracasei and L. casei. The authors should make their point more clear to readers. P2 - what is CheckM? The authors should make a clear comparison 1, how the new assembly is much better than the old one, and 2. to convice the readers that the new assembly is good enough (so the chance of mis-identification again is ver low). P2 - Section 2.1 title -- if there is actually NO sequencing in this work, the word "sequencing" should be removed to avoid misleading. The authors mentioned and used many softwares that are now all known to the readers. The functions/purposes of the softwares (such as SPAdes v3.11.1, CheckM v1.0.11, NxTrim, Benchmarking Universal Single-Copy Orthologs (BUSCO), Roary, FastTree) should be briefly described before the names are mentioned. I don't think readers are all familiar with those softwares. P3 and P4. Please explain "ANI" and "AAI"? How do their values lead to the conclusion to place the re-assembled genome under L. paracasei species? This should be clearly explained. Also, how confident is this classification? What is the probability that a mis-classification could happen? Table 1: in addition to listing the new classification, the confidence score or probability should be given. Figure 2: Not sure what is the message from the heat map? The authors should state the relevance. P6 and Figure 3: The authors should explain how does the software identify the core genome (243 gene families (8225 genes))? Also Figure 3B is not explained clearly? Perhaps the authors should introduce a few strains not from L. paracasei so the readers can see some variations? P11: the LPXTG motif is unique for L. paracasei species but not in other non-L. paracasei species? what is the message here?
Minor:
In Figure 1, the colors of outside circles (magenta, cyan) are too bright and not obvious. Page 8 lane 18, characterizationof? The figures and tables in the PDF are not correctly placed.
Author Response
To
The Editor in Chief
MDPI-Microorganisms,
We thank the anonymous reviewer. Their comments were highly valuable for providing the best relevant information to the journal readers. We believe this improved the quality of the manuscript immensely. Here is a point to point response to the reviewer’s comments.
We will be happy to answer any other query/concern the reviewer or the editors may have.
With Best Regards
Sincerely,
Dr. Sucheta Tripathy
Principal Scientist, Ramalingaswamy Fellow, Associate Professor (AcSIR)
Structural Biology & Bio-Informatics Division
CSIR-Indian Institute of Chemical Biology
4, Raja S. C. Mullick Road.
Kolkata-700032. INDIA
Ph: 91-33- 24995894
Fax: 91-33-2473-5197
Email: tsucheta@gmail.com
tsucheta@iicb.res.in
Major:
Q1. Use too many tables to explain the result, it seems that you didn't well organize them and just give all your raw data. Please try to organize them and use appropriate figures to show your results. Following up major 2 of last time. In S3 table, it’s better to sort the lists by the column of completeness (CheckM) and completeness BUSCOs (BUSCO), so the readers can compare if the “ranking” of CheckM and BUSCO are similar. In addition, what is the reason to use two methods to evaluate the degree of completeness? Maybe it is enough using only one method?
Our response: Thank you for this comment and this definitely makes sense to combine the outputs. CheckM as a package relies on genome completeness based on the marker genes that are usually present in a single copy. Presence of two genes of the single copy markers flag it as contamination and it is a useful metric for complementing BUSCO output. BUSCO on the hand, has lineage specific core orthologs and indicates whether or not the core genes present in the assembly. So, comparing the two outputs will give valuable insight into the results. This time, as suggested, we have combined the BUSCO and CheckM results in one sheet, as well as provided the raw output for the readers to go back and check for additional information. According to terminologies used in CheckM, contamination level of 100% means each of the marker gene is present one more time. CheckM contamination of 0.87 indicates a very low level of contamination. Contamination is used in combination with strain heterogeneity. A 100% strain heterogeneity means the contaminant is very close to the species in question. A lesser score indicates the divergence of the contaminant from the source organism.
Q2. Within all the heatmaps, some of the rows are the same color but belong to different cluster. Suggest to use a smooth gradient color (not only 9 colors) and change the scale and breaks of the color key, so that the readers can easily recognize the difference between each rows and columns.
Our response: Based on your suggestion we have added smooth gradient color to all the heatmaps presented along.
Q3. In Figure 3B, heatmap is not the appropriate figure, as pie chart is better if you only want to explain which function is more. The core COG copy number should be same in different strains, so there is no need to do heatmap and cluster.
Our response: Thank you for your observation. Based on your suggestion we replaced heatmap with pie chart in Figure 3B.
Q4. In Figure 4, it only needs one figure to explain Ka/Ks ratio.
Our response: We have now kept only one figure (histogram plot) to explain Ka/Ks ratio as per your suggestion.
Q5. In Figure S11B, if you use the data of core genes to draw heatmap, the numbers should be same in these 3 strains. Why they are different color in Lpc37? According to your discussion (page 10 lane 4-16), pie chart may be more appropriate if you only want to explain which function is more.
Our response: We have now corrected the error and replaced heatmap with pie chart in Figure S11B.
Q6. In Figure S12 and Table S15, why most of COG categories affiliated to M, G, V? Not G, X, V (the total number is higher)?
Our response: We apologize for the unintentional mistake. Now it has been corrected to G, X,V under section 3.10. (Marked as green after Table2).
Q7. Is it a work simply to correct the authors' previous work (bad assembly leading to mis-identification of strains)? If so, I am not sure this work (erratum) justifies a new research paper?
Our Response: When we submitted the genome assembly as an announcement paper, we did not carry out any analysis on them. It was just an announcement with the intention to carry out more rigorous analysis later. We have been working on the assembly aspect constantly for the last couple of years trying to improve it. Ever since many new tools have been published and one such tool we have extensively used in this study is CheckM. This particular tool analyzes the assembly for their completeness using several marker genes that are invariably single copy. On the basis of presence of extra copies of those genes, it determines the levels of contamination percentage. And also based on extensive ANI and AAI, we could see that our species is falling in L. paracasei rather than L. casei. So, this is far from an erratum. This is a completely new analysis and additionally, we were wondering if other genomes submitted to Genbank have any issues. So, we carried out extensive comparative genomics analysis. It will not be appropriate to assume that this work is just a correction of the previous assembly and an erratum will suffice to describe the quantum of information we are providing here. After reassembly we did large scale comparative genomics, phylogenomics and pangenome analysis (n=75) to decipher important biological properties like probiotic traits. During the recent submission of Lbs2 strain we reported this to NCBI and NCBI acknowledged it. Recently for the other new genome submission from our group we came to know that NCBI is also now doing a QC on the submitted genomes with CheckM.
Q8. P1 - The authors claim that in NCBI, strains of L. casei species are misclassified and these strains are needed to be placed under L. paracasei species. I am not sure how solid the evidences are there to call "misclassified" and what are the alternative evidences that NCBI established at the first place. The authors should present clear and fair arguments before they claim so called "misclassified". This is particularly important as it is the foundation to the whole manuscript.
Our Response: Thank you very much for raising this question. After seeing the ANI (Average Nucleotide Identity) and AAI (Average Amino acid Identity) results (Supplementary Figure S1 and Supplementary Table S1) initially we are in a bit of confusion whether we should report this or not. Later we found this kind of work has been reported in few other publications and they are already cited in the main manuscript. Some of the excerpts are as below: “According to these results, only 5 of 36 genomes annotated as L. casei in the NCBI database are, in fact, genuine members of the species L. casei. The rest should be classified as L. paracasei instead, making L. casei the least sequenced species within the L. casei group”[3] from the manuscript entitled “Large-Scale Phylogenomics of the Lactobacillus casei Group Highlights Taxonomic Inconsistencies and Reveals Novel Clade-Associated Features” [3]. Here we have been trying to make an attempt to clean up the repository, so that mis-information is not propagated.
Q9.P2 top 2 paragraphs -- Not sure how those features are related to the comparative genomics between L. paracasei and L. casei. The authors should make their point more clear to readers.
Our Response: We did comparative genomics among the L. paracasei strains (n=75) and took L. casei strains to find out only taxonomic inconsistency. Therefore, we have written a little bit broader introduction, which is not restricted to only L.casei and L. paracasei species. In introduction, in first paragraph we are talking about natural habitat of Lactobacilli. In second paragraph we are talking about about taxonomic inconsistency and in third and fourth paragraph about important probiotic properties such as sortase enzyme and their action on proteins caring LPXTG motif, production of antimicrobial components like bacteriocins, carbohydrate and vitamin metabolism that are crucial for host energy metabolism.
Q10. P2 - what is CheckM? The authors should make a clear comparison 1, how the new assembly is much better than the old one, and 2. to convice the readers that the new assembly is good enough (so the chance of mis-identification again is ver low).
Our Response: CheckM is a set of tools for assessing the quality of genomes recovered from isolates, single cells, or metagenomes. It provides robust estimates of genome completeness and contamination by using collocated sets of genes that are ubiquitous and single-copy within a phylogenetic lineage (parks et. al., 2015).
CheckM analysis of older version of Lbs2 genome now added to the Supplementary Table S3 and we have marked it in blue in the row number 86. For the readers it will be now easier to understand the assembly quality difference between older and newer version of Lbs2 genome. This part already discussed in result and discussion part under section 3.1. and now marked as green. We also performed ANI and AAI analysis of NCBI defined L. paracasei genomes (n=74) (including reference and representative genome) with newer version of Lbs2 genome (Supplementary Table S1 and Supplementary Figure S1) and we observed identity value for Lbs2 strain is >95% across the strains (n=74). These results again confirmed that newer version of genome quality is good enough and there is no scope for further mis-identification. Moreover newer version of the Lbs2 genome now available at Refseq (where Refseq has quality assessed) along with Genbank whereas older version was available at Genbank only (now removed).
Q11.P2 - Section 2.1 title -- if there is actually NO sequencing in this work, the word "sequencing" should be removed to avoid misleading.
Our Response: The word sequencing now removed from Section 2.1 title. Now it is part of Section 2.2 title (Marked in green).
Q12. The authors mentioned and used many softwares that are now all known to the readers. The functions/purposes of the softwares (such as SPAdes v3.11.1, CheckM v1.0.11, NxTrim, Benchmarking Universal Single-Copy Orthologs (BUSCO), Roary, FastTree) should be briefly described before the names are mentioned. I don't think readers are all familiar with those softwares.
Our Response: We have now discussed the softwares in brief in section 2.1 (also marked in green).
Q13. P3 and P4. Please explain "ANI" and "AAI"? How do their values lead to the conclusion to place the re-assembled genome under L. paracasei species? This should be clearly explained. Also, how confident is this classification? What is the probability that a mis-classification could happen? Table 1: in addition to listing the new classification, the confidence score or probability should be given.
Our Response: ANI or Average Nucleotide Identity and AAI or Average Amino Acid Identity is used for species boundary demarcation. If the ANI and AAI values are >95%, then during multiple genome comparison these genomes are considered to be a part of same species. This threshold value i.e> 95% identity was set up by Goris et.al (for ANI) and Konstantinidis et.al (for AAI). This has already been discussed in section 3.1 (discussed in page number 4 line number 22-32), (supplementary Table S1 and Table S2) (Supplementary Figure S1 and Figure S2). Here, confidence of classification or probability of mis-classification of new assembled genome depends on only ANI and AAI values from multiple genome (established species genome) comparison. So, confidence scores can’t be assigned to this. Table 1 is a summary of the outcome of the ANI and AAI analysis. We have provided Table S2 for reference with raw values.
Q14. Figure 2: Not sure what is the message from the heat map? The authors should state the relevance.
Our Response: Heatmap representing overall distribution of comparative annotation of 75 L. paracasei genomes including Lbs2 strain into 21 COG categories. This heatmap also explaining overall metabolic profile across the 75 strains. Carbohydrate metabolism (G) of Lactobacilliis important for its fitness in the surrounding environment which is discussed in the result section. This is corroborated with this heatmap where carbohydrate metabolism (G) associated COG was found to be highest compared to other COG categories across the 75 strains. Apart from that, this heatmap is also used as a part of phylogenetic analysis. Furthermore, we must say that classification of proteins encoded in sequenced genomes is critical for making the genome sequences maximally useful for functional and evolutionary studies.
Q15. P6 and Figure 3: The authors should explain how does the software identify the core genome (243 gene families (18225 genes))?
Our Response: Roary pipeline uses in built CD-HIT clustering algorithm which clustered the protein sequences with a sequence identity of 100% and a matching length of 100%. If one sequence is identical along its entire length to other orthologous counterparts in other species/strains, then it is said to be a core gene. In our analysis we got 243 gene clusters/families or 243x75 = 18225 genes across the 75 L. paracasei strains.
Q16. Also Figure 3B is not explained clearly? Perhaps the authors should introduce a few strains not from L. paracasei so the readers can see some variations?
Our Response: Figure 3B heatmap now replaced with pie chart as you suggested in Q3 earlier. We are hoping the interpretation is much clearer now.
Q17. P11: the LPXTG motif is unique for L. paracasei species but not in other non-L. paracasei species? what is the message here?
Our Response: We have thoroughly checked the manuscript but could not locate where it states that LPXTG is an unique motif. We on the other hand have talked about the uniqueness of thiamin biosynthetic gene that was found in Lbs2 genome after pangenome analysis (Supplementary Table S7). Moreover, protein containing LPXTG motif is a key for the interaction of bacteria with mucus layer. This motif can be present in both intestinal pathogenic or non pathogenic bacteria.
Minor:
Q1. In Figure 1, the colors of outside circles (magenta, cyan) are too bright and not obvious.
Our Response: Now brightness reduced.
Q2. Page 8 lane 18, characterizationof?
Our Response: Now it has been corrected (marked as green).
Q3. The figures and tables in the PDF are not correctly placed
Our Response: When we convert the .docx file to pdf some format issue arises. Therefore we request to the editorial office staff to insert the figure and tables in proper position.
Round 3
Reviewer 2 Report
Follow up major Q2 of last time. You only change the color, but this is not “smooth gradient”. Please check this: https://www.visualcinnamon.com/2016/05/smooth-color-legend-d3-svg-gradient.html The responses to Q7, Q8, Q10, Q13 and Q15 should be included in the main text.
Author Response
Q1. Follow up major Q2 of last time. You only change the color, but this is not “smooth gradient”. Please check this: https://www.visualcinnamon.com/2016/05/smooth-color-legend-d3-svg-gradient.html
Our response: Based on your suggestion we have added smooth gradient color to all the heatmaps (Figure 2, Figure S1, Figure S2, Figure S6 and Figure S9) with an R package, called pheatmap (earlier we have used gplot package). Now each row and column differences are visible. This pheatmap package use d3.js in backend for smooth gradient.
Q2. The responses to Q7, Q8, Q10, Q13 and Q15 should be included in the main text.
Our response: We are now highlighting the position in manuscript where earlier mentioned responses are included. Details are given below.
Reviewer asked in Q7 about the justification of this study whether it is simple erratum or new work, response to this question was already made in earlier round of revision (round 2). The justification was already there in the Introduction in page no. 2 between line number 18 and 28. Similar reasoning is also there under conclusion section between line numbers 11-26 in Page 15. [all lines are marked as blue]
For Q8- please check Page1 line no. 37-41 (under section 1.) [all lines are marked as blue]
For Q10- please check Page 2 line no. 38-40 (under section 2.1) and Page 5 line no. 2-5 (under section 3.1) [all lines are marked as blue]
For Q13- please check Page 4 line no. 22-35 (under section 3.1) [all lines are marked as blue]
For Q15- please check Page 7 line no. 8-12 (under section 3.2) [all lines are marked as blue]